# c-MYC-Driven Polyamine Metabolism in Ovarian Cancer: From Pathogenesis to Early Detection and Therapy

**DOI:** 10.3390/cancers15030623

**Published:** 2023-01-19

**Authors:** Yihui Chen, Ricardo A. León-Letelier, Ali Hussein Abdel Sater, Jody Vykoukal, Jennifer B. Dennison, Samir Hanash, Johannes F. Fahrmann

**Affiliations:** Department of Clinical Cancer Prevention, The University of Texas MD Anderson Cancer Center, Houston, TX 77030, USA

**Keywords:** ovarian cancer, MYC, polyamine, early detection, therapy

## Abstract

**Simple Summary:**

This review informs on how the MYC signaling is dysregulated and participates in ovarian cancer progression, and the unmet challenge to directly target MYC for ovarian cancer treatment. Therefore, we proposed to alternatively target essential downstream polyamine metabolism pathway of MYC. In this review we include the metabolism of polyamine, the regulation of polyamine metabolism by MYC signaling, the utility of polyamine as therapeutic targets and cancer biomarkers.

**Abstract:**

c-MYC and its paralogues MYCN and MYCL are among the most frequently amplified and/or overexpressed oncoproteins in ovarian cancer. c-MYC plays a key role in promoting ovarian cancer initiation and progression. The polyamine pathway is a bona fide target of c-MYC signaling, and polyamine metabolism is strongly intertwined with ovarian malignancy. Targeting of the polyamine pathway via small molecule inhibitors has garnered considerable attention as a therapeutic strategy for ovarian cancer. Herein, we discuss the involvement of c-MYC signaling and that of its paralogues in promoting ovarian cancer tumorigenesis. We highlight the potential of targeting c-MYC-driven polyamine metabolism for the treatment of ovarian cancers and the utility of polyamine signatures in biofluids for early detection applications.

## 1. Introduction

Ovarian cancer remains among the leading causes of cancer-related mortality among women. The 5-year overall survival rate for ovarian cancer is approximately 50% and decreased to nearly 30% for patients with distant metastasis [1]. It is estimated that there will be 20,000 new ovarian cancer cases and 13,000 deaths due to ovarian cancer in the United States in 2022 [1].

The *MYC* family of transcription factors *c-MYC, MYCN,* and *MYCL* are closely associated with ovarian and other cancers through amplification or overexpression [2,3,4,5,6,7]. MYC family members contain several conserved regions, including the MYC homology boxes (MBs), transactivation domain (TAD), basic region (BR), and helix–loop–helix–leucine zipper (HLH-LZ) domain [8]. Several c-MYC interacting proteins have been identified, including MYC-associated factor X (MAX) [9], WD repeat-containing protein 5 (WDR5) [10], TATA box-binding protein (TBP) [11], and bridging integrator 1 (BIN1) [12], whereas Aurora kinase A (AURKA) has been shown to interact with MYCN through MBI and flanking regions on MYCN [13]. These interactors enable MYC family members to function as transcription factors to promote oncogenesis. The c-MYC heterodimer can also repress gene expression via binding to the transcription factors MYC-interaction zinc-finger protein 1 (Miz1) and specificity protein 1 (SP1) [14,15].

c-MYC expression and function are tightly regulated at both the transcriptional level by a variety of transcriptional regulatory motifs within its proximal promotor region [16], as well as at the protein level through post-translational modifications, namely phosphorylation at various serine and threonine residues that promotes proteasomal degradation through the canonical SCF^FBXW7^ ubiquitin ligase pathway [17].

The MYC family oncoproteins, particularly c-MYC, are essential master regulators of metabolic reprogramming in a variety of cancer types, including ovarian cancer [18,19]. Among the c-MYC-regulated metabolic pathways is the polyamine biosynthetic pathway. Polyamines are ubiquitously distributed highly charged small basic molecules that are strongly associated with hyperproliferative diseases and play essential roles in c-MYC-driven malignancies by promoting cell growth and other pro-tumoral activities [20,21]. Cancer cells upregulate polyamine pools either by increasing de novo biosynthesis or enhancing salvaging from the surrounding milieu. Catabolism of polyamines is mediated by spermidine/spermine N1-acetyltranferase (SSAT), which is responsive to intracellular polyamine pools, resulting in the formation of acetylated polyamine derivatives, such as diacetylspermine (DAS), that can be secreted into the surrounding microenvironment and can be detected in peripheral blood, providing potential utility for cancer detection [22]. Targeting dysregulated polyamine metabolism is also being increasingly explored as a therapeutic strategy for various malignancies, including ovarian cancer.

In this review, we discuss the significance of c-MYC and its paralogues in ovarian cancer. We additionally highlight opportunities for exploiting c-MYC-driven polyamine metabolism for early detection and therapeutic applications in ovarian cancer.

## 2. MYC Family Members and Ovarian Cancer

### 2.1. Genomic Aberrations and Overexpression of c-MYC in Ovarian Cancer

*MYC* is located on chromosome 8q24 [23,24], a commonly amplified region in ovarian carcinomas [25]. Genomic alterations in MYC family members are prevalent in ovarian cancer [2,4]. c-MYC copy-number amplifications have been reported in up to 50% of ovarian carcinomas, whereas MYCL copy-number amplifications and gene overexpression have been reported in upwards of 21% of ovarian carcinomas [6,26,27,28,29,30,31,32,33,34,35,36,37]. *MYCN* amplification has also been frequently found in ovarian cancers [38].

*c-MYC* transcript and protein levels are higher in ovarian tumors compared to normal tissues, and c-MYC overexpression is associated with more aggressive disease [39,40]. *c-MYC* gene expression tends to correlate with *c-MYC* copy-number amplification [41,42,43], although these are not mutual. *MYCL* and *MYCN* gene transcripts also are overexpressed in some ovarian cancers [38,39,44].

c-MYC transcriptional activity is, in part, modulated post-translationally through phosphorylation and proteasomal degradation [45,46]. Phosphorylation at serine 62 (Ser62) by mitogenic signaling increases c-MYC stability [47]. Studies have demonstrated that mutations at the threonine 58 residue result in constitutive Ser62 phosphorylation [48], leading to c-MYC protein overexpression and activation. Cancer cells also maintain Ser62 phosphorylation of c-MYC by downregulating protein phosphatase 2A (PP2A), a p-Ser62 targeting phosphatase [49,50,51,52]. In ovarian cancer, proto-oncogene serine/threonine-protein kinase Pim-1 interacts with c-MYC and mediates its Ser62 phosphorylation, promoting cancer cell proliferation [53]. F-Box protein 31 (FBOX31), which is suppressed by c-MYC, facilitates the ubiquitination and degradation of c-MYC independent of its phosphorylation status [54].

### 2.2. Prognostic Value of c-MYC in Ovarian Cancers

Several studies have assessed the prognostic value of c-MYC copy-number amplification as well as mRNA expression and protein levels in ovarian cancer. Wang found that ovarian cancer patients with *c-MYC* copy-number amplifications had poorer overall survival compared to those with wild-type *c-MYC* [29]. Similarly, Katsaros et al. reported that *c-MYC* copy-number amplification with high p185/p21 in ovarian tumors was associated with shorter patient survival [28]. Elevated tumor c-MYC mRNA levels were prognostic for poor disease-free survival and worse overall survival in The Cancer Genome Atlas (TCGA) ovarian cancer dataset [55]. MYC protein levels and localization account for its transcriptional activity. Two independent studies revealed positive correlations between the c-MYC protein levels and sub-cellular localization with tumor size and tumor classification, respectively [56,57]. Ning et al. found that enrichment of c-MYC in the nucleus in early-stage ovarian cancer correlated with shorter overall survival [43]. Yamamoto et al. showed that phosphorylation of c-MYC at Ser62 was prognostic for poor overall survival [42]. The above-mentioned studies suggest a prognostic role of c-MYC. However, it should be noted that several studies have reported non-significant associations between c-MYC copy numbers as well as mRNA and protein levels in ovarian tumors with survival outcomes [30,34,39,42,58,59,60]

Collectively, these findings emphasize the need to integrate the multidimensional data incorporating the complexity of c-MYC signaling at the protein level, localization, as well as PTM status, to define the relationship between c-MYC and ovarian cancer.

### 2.3. Therapeutic Targeting of MYC in Ovarian Cancers

Oncogenic c-MYC and its paralogues orchestrate tumor-promoting signaling through reprograming of cell growth, survival, metabolism, and shaping the tumor microenvironment (TME) and tumor immunity [18,61,62]. Many cancers, including ovarian cancer, are addicted to c-MYC signaling [62,63,64,65], which has served as the justification for targeting c-MYC [66,67,68,69]. Early investigations demonstrated anti-ovarian cancer efficacy by targeting c-MYC using triplex-forming and liposomal phosphorothioate oligonucleotides in vitro [70,71]. A small molecule 10058-F4 blocks c-MYC/MAX heterodimerization in ovarian cancer cells and thus induces cell cycle arrest and apoptosis and attenuates glutamine uptake, an essential nutrient for cancer cells [72,73]. Another molecule, JQ1, targeting c-MYC and BRD4, suppresses ovarian cancer cell proliferation and induces apoptosis [74,75,76]. Concomitant upregulation of FAK and c-MYC in HGSOC implies a co-targeting strategy. Dual targeting of FAK and MYC by VS-6063 and JQ1 resulted in cell cycle arrest and cell death of ovarian cancer cells in vitro [77]. Moreover, c-MYC amplified chemotherapy-resistant ovarian cancer cells were demonstrated to be highly sensitive to combinational JQ1 and GS-626510, a bromodomain and extra-terminal motif inhibitor [75]. Indirect inhibition of c-MYC by the CDK7 and CDK12/13 inhibitor THZ1 repressed tumor growth in patient-derived xenograft models of ovarian cancer [65]. Dual inhibition of PARP and CDK4/6 suppressed ovarian cancer cell growth in vitro and tumor growth in vivo in a c-MYC-dependent manner [78]. The insulin-like growth factor II mRNA-binding protein 1 (IGF2BP1/IMP1) inhibitor BTYNB destabilizes *c-MYC* mRNA via disrupting IMP1 binding to *c-MYC* mRNA, thereby repressing ovarian cancer cell proliferation [79]. In addition, targeting c-MYC also sensitizes ovarian cancer cells to chemotherapy [55,75,76].

Despite the above-mentioned studies, targeting c-MYC and its paralogues has remained challenging due to protein and peptides inhibitors undergoing rapid proteolytic degradation, as well as issues related to the low membrane and cell permeability, low oral bioavailability, high clearance, and poor tissue distribution [80]. Currently, there are no drugs clinically available for targeting c-MYC in human cancers. An alternative strategy is to target one or several downstream signaling pathways of c-MYC.

## 3. MYC and Polyamine Regulation

As an oncogenic transcription factor, c-MYC and its paralogues transcriptionally regulate hundreds of genes that impact tumor metabolism and enable tumor progression [3,19,62,81]. c-MYC is involved in nutrient sensing and regulates glucose, glutamine, and lipids metabolism by upregulating glucose transporter 1, pyruvate kinase isoenzymes M1 and M2, and lactate dehydrogenase A [19]. Among other c-MYC-regulated metabolic pathways is the polyamine metabolism pathway, which plays critical tumor-promoting roles [20,82,83,84].

Polyamines are small polycationic molecules present in nearly all living organisms. The three major mammalian polyamines (putrescine, spermidine, spermine) play essential roles in cell proliferation, and functionality and polyamines are commonly linked to hyperproliferative disorders, including cancer. Polyamine content is usually higher in rapidly growing cells and tissues and is upregulated by regenerative and growth-promoting hormonal stimuli [85,86,87]. Inhibition of polyamine biosynthesis induces cytostasis, which is reversible by the supplementation of exogenous putrescine or spermidine [88]. The best-known molecular function of spermidine is serving as the aminobutyl group donor for the hypusination of eIF5A, which is essential for the activation of this translation factor that contributes to transcription, mRNA turnover, nucleocytoplasmic transport, and apoptosis [89].

### 3.1. Biosynthesis of Polyamines

Putrescine, spermidine, and spermine are the only three polyamines that can be synthesized de novo in mammalian cells [21]. Putrescine is the central intermediate of polyamine biosynthesis. Ornithine decarboxylase 1 (ODC1), in the presence of cofactor PLP, produces putrescine through the decarboxylation of ornithine (Orn), a metabolic product of arginase [90,91]. ODC1 is the rate-limiting step in putrescine production and is a bona fide target of oncogenic c-MYC (Figure 1).

The aminopropyl moiety is required to form higher polyamine molecules, such as spermidine and spermine, from putrescine [20,21]. S-adenosylmethionine decarboxylase (AdoMetDC, encoded by *AMD1*) converts S-adenosylmethionine (AdoMet) into decarboxylated AdoMet (dcAdoMet), the aminopropyl group donor [92]. Putrescine stimulates the processing of mammalian AdoMetDC and enhances its activity (Figure 1) [92,93]. Therefore, putrescine facilitates its conversion resulting in higher levels of spermidine and spermine than putrescine in mammalian cells and tissues.

Spermidine and spermine are the most pronounced non-acetylated polyamines in mammals. Aminopropyltransferases, namely spermidine synthase (SRM) and spermine synthase (SMS) are responsible for the formation of these higher polyamines. SRM transfers the aminopropyl moiety to putrescine to generate spermidine, and SMS adds another aminopropyl group to spermidine from spermine (Figure 1) [94]. Despite their similar aminopropyl transferring activity, SRM and SMS are distinct enzymes with strict substrate specificity [95,96]. This difference prevents the cross-reaction between these enzymes and their substrates.

### 3.2. Exogenous Sources of Polyamines

Although mammalian cells have an intact pathway for de novo biosynthesis of polyamines, exogenous sources are additional means to maintain polyamine homeostasis (Figure 1) [97,98,99,100]. Polyamines are found in all types of food and provide hundreds of micromoles of polyamines to the gut lumen daily [101,102,103]. For instance, agmatine, which is not synthesized by mammalian cells but is still widely present in mammalian tissues, is a polyamine produced by plants and microorganisms and mainly obtained from the diet and gut microbiota [21,104]. Sawada et al. suggested that ingested food is the major source of polyamines in the lumen of the upper small bowel in humans [105], while the gut microbiota, to some extent, is responsible for polyamines level mainly in the lower part of the intestine [106,107]. A polyamine-rich diet significantly increases levels of circulating polyamines in mouse and human [108,109].

Luminal polyamines are rapidly absorbed in the upper intestine [110]. In addition to passive diffusion [111,112], endocytosis and solute carrier-dependent mechanism also participate in polyamine uptake [113]. A recent study demonstrated that ingestion of exogenous putrescine is mediated by a caveolin-1 and NOS2-dependent process, while the solute carrier transporter SLC3A2 imports putrescine from the polyamine-rich lumen [114]. Colonic mucosa is also involved in transporting polyamines from the colonic lumen into the bloodstream [115]. It is becoming increasingly evident that exogenous polyamines affect whole-body polyamine levels and have an impact on a variety of activities in health and diseases [97,98,101,103,116,117]. Understanding the importance of exogenous polyamines requires further investigations.

### 3.3. Mechanisms of Extracellular Polyamines Uptake

Importing polyamines from the extracellular compartment has been shown to be of equal importance as de novo polyamine biosynthesis. Several studies have demonstrated the diverse activities of extracellular polyamines [118,119,120,121]. However, the mechanisms driving polyamine uptake at the molecular level remain incomplete. A study suggested that chemicals modulating the membrane potential affect the polyamine import system, and protein kinase C inhibits polyamine uptake [122]. Heparan sulfate is also implicated in polyamine import, which can be blocked by the single-chain fragment of an anti-heparan sulfate antibody [123]. Glypican-1 (GPC-1), along with other glypicans, is a family of heparan sulfate proteoglycans [124]. The Heparan sulfate side chains of recycling GPC-1 sequester spermine for its uptake [125]. By using a fluorescent polyamine probe, Soulet et al. demonstrated the internalization and localization of polyamines in discrete vesicles [126]. Another study provided a supportive observation of caveolin-1-dependent endocytosis-mediated polyamine uptake [127]. A previous study suggested that the solute carrier transporter SLC3A2 is an exporter of polyamines [128], while an independent study demonstrated that SLC3A2 also mediates polyamine import [129]. To this end, Uemura et al. demonstrated that SLC3A2 is a bidirectional polyamines transporter that can either import or export polyamines depending on the concentration gradient [114]. In addition, the ATP13A2 and ATP13A3 ATPases are also implicated in polyamine uptake, although the biochemical mechanism is not fully understood [130,131,132]. Multiple mechanisms may cooperate to regulate polyamines import, and their involvement could be tissue-specific and context-dependent.

The export of polyamines is important to maintain polyamine homeostasis. This process is well-studied in bacteria, yeast and protozoa [133,134,135,136,137,138,139,140,141]. Although the excretion of these compounds is also reported in mammalian cells [142,143,144,145,146], the underlying molecular mechanism remains elusive. To date, SLC3A2 is the only well-studied exporter of polyamines [114,128]. The solute carrier 18B1 (SLC18B1), a member of the vesicular amine transporter family, mediates the exocytosis of polyamines from mast cells [147], and another study reported the requirement of SLC18B1 to maintain polyamine pool in the brain [148]. There are many other transporters that have been implicated in polyamine effluxion [149], but extensive research is required to characterize these polyamines’ secretory pathways.

### 3.4. Catabolism of Polyamines

Polyamines are interconvertible, and several enzymes involved in these processes have been identified (Figure 1) [150,151,152,153]. Spermine oxidase (SMOX) selectively catalyzes spermine into spermidine and 3-aminopropanaldehyde [151]. Several SMOX isoforms resulting from alternative splicing have also been identified, although the functional consequence of these distinct isoforms is largely unknown. These isoforms share similar catalytic activities but differ in their localization [154,155,156,157,158]. There is also an alternative pathway through which spermine is converted into spermidine and spermidine into putrescine [153]. In this pathway, a less specific acetyltransferase SSAT adds acetyl groups to either spermine or spermidine [159,160]. The N1-acetylspermine (AcSpm) or N1-acetylspermidine (AcSpmd) is then catabolized into spermidine or putrescine, respectively, in the presence of the acetylpolyamine oxidase (PAOX) [156,161,162,163]. PAOX prefers to oxidize acetylated polyamines and exhibits negligible affinity towards spermidine and spermine [157]. The SMOX and SSAT/PAOX pathways work against the polyamine biosynthesis reactions to degrade the higher polyamines into putrescine, controlling the polyamines dynamics. Putrescine can be oxidized by the diamine oxidase [164,165] or excreted in poorly characterized manners [166]. In addition, the higher polyamines, as well as their acetylated derivatives, can also be secreted from cells [166,167,168,169].

### 3.5. Regulation of Polyamines by c-MYC

Polyamine metabolism is exquisitely regulated by c-MYC signaling [83]. The first piece of evidence for the interplay between polyamine and the *c-MYC* oncogene was revealed with the observation that *ODC1* is a transcriptional target of c-MYC [170]. c-MYC-induced growth factor-independent transcription results in increased ODC1 protein expression and enhanced enzymatic activity, resulting in cell cycle progression and transformation [171,172,173,174,175]. A single nucleotide A/G polymorphism at position +317 in the human *ODC1* gene relative to transcription initiation is located between the two E-boxes and may affect c-MYC binding and, therefore, *ODC1* expression [176,177]. Moreover, ODC1 is a critical determinant of *c-MYC* oncogenesis and a potential therapeutic target in MYC-driven tumors [178].

In addition to *ODC1*, other genes in the polyamine metabolism pathway regulated by c-MYC include *SRM*, *SMS*, *AMD1*, and *OAZ2,* an ornithine decarboxylase antizyme [178,179,180]. c-MYC also indirectly modulates AdoMetDC/AMD1 expression by promoting mTOR activation [181,182,183], which stabilizes AdoMetDC [184]. Regarding polyamine catabolism, c-MYC has been demonstrated to repress the expression of *SSAT* and *SMOX* [178,179], thus maintaining high cellular polyamine levels.

There is no direct evidence for c-MYC-regulated polyamine transport, but some indirect evidence suggests that c-MYC promotes polyamine uptake. However, the polyamine transporter *SLC3A2* is a direct transcriptional target of c-MYC [185,186].

## 4. Polyamines as Therapeutic Targets in MYC-Driven Ovarian Cancer

Polyamines act as critical regulators of tumor initiation, and progression and dysregulation of polyamine homeostasis are prevalent in various human cancers [20,187,188]. The potential application of targeting polyamine metabolism in cancer therapy remains an active area of intensive investigation in a variety of cancers, including ovarian cancer [20,187]. An overview of small molecular inhibitors targeting various aspects of polyamine metabolism is provided in Figure 1.

### 4.1. Targeting Polyamine Metabolism and Transport for Ovarian Cancer Treatment

The ODC1 inhibitor eflornithine (also known as α-difluoromethylornithine or DFMO) was developed in 1978 [189]. Early studies have investigated the cytotoxicity of DFMO in ovarian cancer cells [190], and a recent study demonstrated that DFMO induces apoptosis in ovarian cancer cells by regulating AP-1 signaling [188]. In addition to a stand-alone treatment modality, researchers have also explored the effect of DFMO in combination with other drugs. Together with the DNA methyl transferase inhibitor 5-azacytidine, DFMO increased the accumulation of anti-tumor M1 macrophages and reversed the immunosuppressive TME, promoting tumoricidal immune responses and prolonging the survival of mice with ovarian cancer [191]. In another study, DFMO synergistically enhanced the cytotoxicity of PARP inhibitors toward ovarian cancer cells in vitro [192].

AdoMetDC is essential to produce the aminopropyl moiety donor for the biosynthesis of polyamines. Targeting AdoMetDC can therefore interrupt polyamine homeostasis in cancer cells resulting in anti-cancer effects. Several AdoMetDC inhibitors have been developed, including methylglyoxal bis(guanylhydrazone) (MGBG)[193], 4-amidinoindan-1-one 2′amidinohydrazone (SAM486A) [194,195,196], as well as 5′(((z)-4-amino-20butenyl)methylamino)-5′deoxyadenosine (AbeAdo) and its 8-methyl derivative (Genz-644131) [187,193]. Evaluation of the anti-cancer efficacy of these small molecule inhibitors for the treatment of ovarian cancer has been limited, although earlier studies demonstrated that DFMO, in combination with AbeAdo, promoted G1 arrest in OVCAR-3 ovarian cancer cells [197].

The anti-cancer effects of small molecule inhibitors targeting de novo polyamine biosynthesis may be attenuated due to increased scavenging of extracellular polyamines by cancer cells. AMXT 1501 (D-Lys(C_16_acyl)-Spm) exhibited extraordinary cytotoxicity towards a variety of cancer types, including ovarian cancer, by interfering with the extracellular polyamine transport apparatus [187]. Moreover, the combination of DFMO and AMXT 1501 treatment also yielded exceptional tumoricidal efficacy [187]. A phase I clinical study has recently been initiated to assess the safety and dose level of AMXT 1501 alone and in combination with DFMO for the treatment of solid cancers, including ovarian cancer (NCT05500508). F14512, another polyamine transport blocker, demonstrated potent anti-proliferative and pro-apoptotic activities in vitro and in a mouse ovarian tumor model. Notably, the anti-cancer effects of F14512 were most profound in ovarian cancer cells that exhibited highly active polyamine transport systems [198]. A phase I clinical study using F14512 was conducted in patients with platinum-refractory or resistant ovarian cancer. However, this study was discontinued due to the occurrence of grade 4 neutropenia [199].

### 4.2. Exploiting Synthetic Polyamine Analogues to Deplete Polyamine Pools in Ovarian Cancer Cells for Anti-Cancer Treatment

Initial attempts have led to the generation of several analogs that are structurally similar to natural polyamines, and several additional analogs have gradually been introduced with an advanced understanding of polyamine metabolism and synthesis methods (Figure 1) [200,201,202]. The representative polyamine analogs that have been intensively studied in cancers include N^1^,N^11^-bis(ethyl)norspermine (BENSpm, also known as DENSpm), N^1^,N^12^-bis(ethyl)spermine (BESpm), N^1^,N^12^-bis(ethyl)-cis-6,7-dehydrospermine (PG-11047), and diethyl dihydroxyhomospermine (SBP-101). Mechanistic studies reveal that BENSpm and BESpm not only interfere with polyamine biosynthesis and transport but also drastically induce the polyamine catabolic enzymes SSAT and SMOX, resulting in the depletion of cellular polyamines and cancer cell death [203,204,205]. Intraperitoneal administration of BENSpm into A121 ovarian carcinoma tumor-bearing mice resulted in pronounced anti-cancer effects, with 40% of tumor-bearing mice being tumor-free following the intervention [206]. Mechanistically, BESpm treatment depleted ovarian cancer cell polyamine pools, partially through induction of SSAT, resulting in cancer cell death [207]. Furthermore, studies using ovarian cancer cell lines demonstrated that the combination of BENSpm and BESpm with chemotherapeutic drugs results in improved anti-cancer effects compared to chemotherapy alone [208,209,210]. SBP-101, unlike other polyamine analogs, had modest induction of polyamine catabolism but robustly repressed the activity of ODC1 in ovarian cancer cells [211]. More recently, a preclinical study reported that, through modulating polyamine metabolism and immunosuppressive microenvironment, SBP-101 marked prolonged the median survival of mice in a VDID8^+^ murine ovarian cancer model [212].

Early studies revealed that platinum-based drugs and polyamine analogs induced SSAT expression in the cisplatin-sensitive, but not the cisplatin-resistant, human ovarian cancer cells [213,214]. Other studies further demonstrated that treatment of ovarian cancer cells with cisplatin or oxaliplatin in combination with DENSpm results in enhanced anti-proliferative effects via enhancing SSAT expression [208,209,215]. These findings are consistent with a previous study that also reported increased sensitivity of ovarian cancer cells to cisplatin when SSAT1 was transiently overexpressed [216]. Notably, in these studies, other polyamines catabolism enzymes SSAT2, SMOX and PAOX were also induced, while ODC1 was downregulated [209,215].

The folate cycle inhibitor-induced SSAT expression results in the depletion of cellular polyamines with subsequent production of reactive oxygen species and cell death in ovarian cancer cell lines [210,217]. Moreover, folate cycle inhibitors, when combined with small molecular inhibitors of polyamine metabolism, were demonstrated to elicit synergistic cytotoxicity toward ovarian cancer cells [210]. This combinational treatment also increased chemosensitivity in drug-resistant ovarian cancer cells [210,217].

## 5. Utility of Polyamines as Biomarkers for Early Detection of Ovarian Cancer

Polyamines exert potent tumor-promoting activities, and their homeostasis is often dysregulated in cancers with increased biosynthesis and excretion, thus providing opportunities for early detection applications. Indeed, polyamines and their metabolites are reported as biomarkers for various cancers, including ovarian cancer (Table 1) [22,218,219,220].

### 5.1. Urinary Polyamines and Their Acetylated Derivates as Biomarkers for Early Detection of Ovarian Cancer

Increased urinary excretion of putrescine, spermidine, and spermine has been observed in ovarian cancer cases [221,222,226,227]. Putrescine, spermidine, and spermine are part of a urine-based biomarker test developed by Waalkes et al. for the detection of ovarian cancer [223]. In their study, the frequency of the three polyamines was markedly elevated in the urine of cases with advanced ovarian cancer compared to disease-free controls (42% vs. 13% for putrescine, 88% vs. 0% for spermidine, and 59% vs. 7% for spermine) [223]. Moreover, the urinary spermidine/creatinine ratio was drastically increased in advanced ovarian cancer patients [223]. Suh et al. evaluated free and acetylated urinary polyamines and found consistent increases in putrescine, spermidine, and spermine levels in the urine of ovarian cancer patients compared with controls [224]. Moreover, N-acetylputrescine (NAcPut) and AcSpmd were also concomitantly elevated in cases compared to healthy controls [224]. In an independent study, 14 urinary-free, mono- and di-acetylpolyamines were measured in patients with ovarian cancer and individuals presenting with benign disease. These studies revealed DAS, among other polyamines, to be able to distinguish malignant from benign masses [225]. Notably, DAS had better sensitivity (86.5%) but lower specificity (65.2%) than CA-125 (75.7% sensitivity, 69.6% specificity) [225].

### 5.2. Plasma Acetylated Polyamines for Risk Prediction of Malignancy for Ovarian Cancer

Polyamines have also been reported to be elevated in the blood of individuals with ovarian cancer [228,229]. Recently, Fahrmann et al. reported a plasma polyamine signature for the early detection of ovarian cancer [22]. Specifically, a logistic regression model based on DAS + N-(3-acetamidopropyl)pyrrolidin-2-one (N3AP) + CA125 was developed for the detection of ovarian cancer. At a >99% specificity threshold, the model yielded a sensitivity of 73.7% sensitivity in an independent validation set for the detection of early-stage ovarian cancers, which was markedly better than CA-125 alone (62.2% sensitivity; McNemar exact test 2-sided P: 0.019) [22]. In an independent study, Irajizad et al. developed a deep learning model based on a panel of seven cancer-associated metabolites, which included DAS, DiAcSpmd, and N3AP, for risk prediction of malignancy among women presenting with ovarian cysts. The 7-marker metabolite panel had an AUC of 0.86 for differentiating early-stage ovarian cancers from benign pelvic masses. The authors further demonstrated that the combination of the 7-marker metabolite panel with the risk of ovarian malignancy algorithm (ROMA) yielded a significantly higher positive predictive value (PPV) compared to ROMA alone (0.68 vs. 0.52) for early-stage ovarian cancer [220].

## 6. Conclusions

c-MYC and its paralogues are among the most frequently amplified and/or overexpressed oncoproteins in ovarian cancer. *MYC* family members contribute to cancer initiation and progression and are regulated at genomic, mRNA, and protein levels. The tumor-promoting polyamine pathway is tightly linked to c-MYC signaling and intertwines with the malignancy of ovarian cancer. Multiple strategies for inhibiting polyamine biosynthesis, metabolism, and transport have been explored in the context of ovarian cancer, with several small molecule inhibitors having been developed to target various aspects of polyamine metabolism. Preclinical investigations have shown considerable promise of polyamine metabolism as a metabolic vulnerability for the anti-cancer treatment of ovarian cancer. Clinical trials targeting polyamine metabolism using small molecule inhibitors for the treatment of solid malignancies, including ovarian cancer, are ongoing. The elevation of polyamines and their acetylated derivates in urine and plasmas of ovarian cancer cases provides additional opportunities for risk prediction and early detection applications and provides a potential means to better select individuals who may best benefit from polyamine targeting therapies.

## Figures and Tables

**Figure 1 cancers-15-00623-f001:**
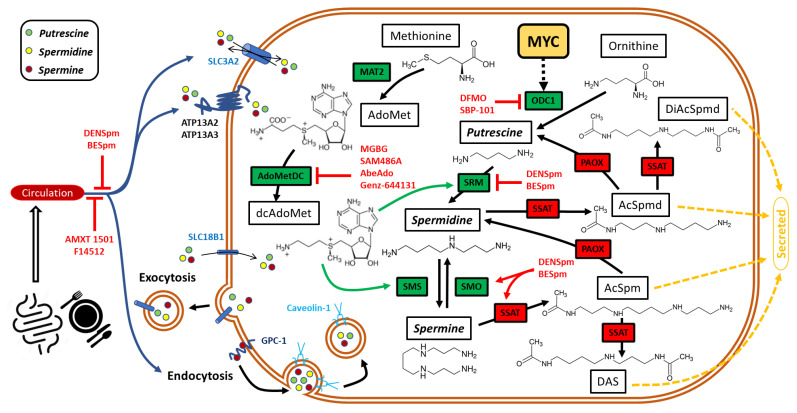
Biosynthesis, metabolism, transport, and targeting of polyamines. (1) Polyamine biosynthesis: ornithine is generated from arginine and converted into putrescine by ornithine decarboxylase (ODC1). Spermidine synthase (SRM) turns putrescine into spermidine, which is further converted into spermine by spermine synthase (SMS). Synthesis of spermidine and spermine requires aminopropyl moiety from decarboxylated S-adenosylmethionine (dcAdoMet), which is converted from S-adenosylmethionine (AdoMet) by S-adenosylmethionine decarboxylase (AdoMetDC). AdoMet is produced from methionine by methionine adenosine transferase (MAT). (2) Polyamine catabolism: spermidine and spermine are decomposed by spermidine/spermine-N1-acetyltransferase (SSAT) into N-acetylspermidine (AcSpmd) and N-acetylspermine (AcSpm), respectively. AcSpmd and AcSpm are further catabolized by acetylpolymine oxidase (PAOX) into spermidine and putrescine, respectively. In addition, spermine oxidase (SMOX) specifically degrades spermine into spermidine. (3) Polyamine transport: exogenous sources of polyamines, including foods and gut microbiota. Uptake of polyamines from the gut may be mediated by caveoline-1- and NOS-2-dependent processes. Solute carrier transporter SLC3A2 imports putrescine. At the cellular level, heparan sulfate proteoglycans glypican-1 (GPC-1) sequesters spermine for its uptake, likely through endocytosis. Caveoline-1-dependent endocytosis mediated polyamine uptake from extracellular compartments. SLC3A2 can either import or export polyamines following their concentration gradient. SLC18B1 promotes polyamine exocytosis in mast cells but is also required for polyamine uptake in the brain. ATP13A2 and ATP13A3 are involved in polyamine internalization via poorly characterized mechanism(s). (4) Targeting polyamine pathways: polyamine analogs or small molecule inhibitors targeting the polyamine metabolism pathway antagonize the tumor-promoting activities of polyamines. Small molecules AMXT 1501, F14512, and polyamine analogs DENSpm and BESpm block the uptake of polyamines by cancer cells. In addition, DENSpm and BESpm also inhibit polyamine synthesis by suppressing SRM or promote polyamine catabolism by upregulating SMOX and SSAT. DFMO and SBP-101 repress ODC, while MGBG, SAM486A, AbeAdo, and Genz-644131 target AMD to downregulate polyamine synthesis.

**Table 1 cancers-15-00623-t001:** Polyamine as biomarkers for ovarian cancer.

Polyamines	Sources of Polyamines	Observations	Reference
Polyamines	Urine	Increased polyamines correlate with clinical status.	[221]
Free and acetylated polyamines	Urine	Free and acetylated polyamines were elevated in cases compared to controls.	[222]
Putrescine Spermidine Spermine	Urine	Polyamines are elevated in patients with progressive diseases; Spermidine/creatinine ratio is increased.	[223]
Putrescine Spermidine Spermine NAcPuT AcSpmd	Urine	Polyamines are drastically elevated in cancer patients	[224]
DAS	Urine	DAS has 65% specificity and 91% sensitivity (AUC 0.82), better than CA-125 (65% specificity, 68% sensitivity, AUC 0.75) and RMI (70% specificity, 68% sensitivity, AUC 0.72)	[225]
DAS N3AP DAS AcSpmd	Plasma	Polyamine signature consisting of DAS and N3AP in combination with CA-125 yields improvement in sensitivity at >99% specificity relative to CA-125 alone (73.7% vs. 62.2%) and can capture 30.4% more cases than CA-125 alone	[22]
DAS N3AP DiAcSpmd	Serum	7MetP yields an AUC of 0.86; 7MetP+ROMA increase AUC from 0.91 (ROMA alone) to 0.93; 7MetP+ROMA has a higher positive predictive value (0.68 vs. 0.52) with improved specificity (0.89 vs. 0.78) compared to ROMA alone.	[220]

DAS: Diacetylspermine, NAcPut: N-acetylputrescine, AcSpmd: N-acetylspermidine, N3AP: N-(3-acetamidopropyl)pyrrolidine-2-one, DiAcSpmd: Diacetylspermidine, 7MetP: 7-marker metabolite panel, ROMA: risk of ovarian malignancy algorithm.

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
