# Peer review of "c-MYC-Driven Polyamine Metabolism in Ovarian Cancer: From Pathogenesis to Early Detection and Therapy"

_cancers, 2023, doi:10.3390/cancers15030623_

Round 1

Reviewer 1 Report (Previous Reviewer 1)

In this review, the authors summarize the importance of the MYC gene family and polyamine metabolism in relation to ovarian carcinoma. in this review, the authors suggest a possible influence on tumor cells in the downstream signaling pathway of c-MYC. Here the authors particularly emphasize the polyamines as a possible target for fighting cancer. Polyamines are regulated exclusively by c-MYC. C-MYC also regulates other genes involved in polyamine catabolism. This effect is known under tamoxifen therapy, which inhibits c-MYC and thus, reduces polyamine catabolism and transport. Further studies show detectable acetylated polyamines in the urine of patients with ovarian carcinoma. This detection option shows improved detection efficiencies than conventional methods. The authors discuss possible points of attack in the polyamine metabolism to combat ovarian carcinoma.

This review raises awareness of a possibly target-oriented field of research in order to be able to obtain more effective early detection options for ovarian carcinoma. I therefore support the publication of this manuscript. I do accept this manuscript.

Reviewer 2 Report (Previous Reviewer 2)

Dear authors,

Thank you for the opportunity to review your manuscript and this new version.

The revised manuscript shows real improvement. It is now well readable and fluently carries a more complete review concerning c-MYC and ovarian cancer.

In my opinion, this review can now bring important subsidies to other professionals interested in continuing to propose research related to c-MYC for diagnosis and targeted therapies.

Best regards

This manuscript is a resubmission of an earlier submission. The following is a list of the peer review reports and author responses from that submission.

Round 1

Reviewer 1 Report

In this review, the authors summarize the importance of the MYC gene family in relation to ovarian carcinoma. They point to conflicting results regarding an association between MYC and clinical courses in ovarian cancer. Studies are presented in which third-party substances nevertheless have an indirect effect on tumor cells and are associated with MYC. With these studies, the authors justify a previously missing approach to inhibit MYC with medication. Therefore, in this review, the authors suggest a possible influence on tumor cells in the downstream signaling pathway of c-MYC. Here the authors particularly emphasize the polyamines as a possible target for fighting cancer. Polyamines are regulated exclusively by c-MYC. C-MYC also regulates other genes involved in polyamine catabolism. This effect is known under tamoxifen therapy, which inhibits c-MYC and thus, also reduces polyamine catabolism and transport. Further studies show detectable acetylated polyamines in the urine of patients with ovarian carcinoma. This detection option shows improved detection efficiencies than conventional methods. The authors discuss possible points of attack in the polyamine metabolism to combat ovarian carcinoma.

Minor:

The general function of the MYC should be explained more clearly in the introduction. The understanding of MYC as a regulatory element in expression and the domains responsible for this should be mentioned in the introduction. This knowledge makes it easier for the reader to better classify the many study results. Genes are often named differently in the text than in the figure.

This review raises awareness of a possibly target-oriented field of research in order to be able to obtain more effective early detection options for ovarian carcinoma. I therefore support the publication of this manuscript. I do accept this manuscript.

Author Response

The general function of the MYC should be explained more clearly in the introduction. The understanding of MYC as a regulatory element in expression and the domains responsible for this should be mentioned in the introduction. This knowledge makes it easier for the reader to better classify the many study results.

Response: We have expanded the introduction section of the revised manuscript to now include information regard MYC domains and interacting partners. 

Genes are often named differently in the text than in the figure.

Response: We have now modified the figure to main consistency with terminology used in the primary text.

This review raises awareness of a possibly target-oriented field of research in order to be able to obtain more effective early detection options for ovarian carcinoma. I therefore support the publication of this manuscript. I do accept this manuscript.

Response: We thank the Reviewer for their support of our Review.

Reviewer 2 Report

Dear authors, I review your review very carefully but I have a few comments to make:

Comments:

It would be useful for the reader to keep the same form of abbreviation. In your text we find this: Line 9: MYC family members, including c-MYC, MYCN, MYCL and Line 29 : The MYC family of transcription factors C-MYC, N-MYC and L-MYC. In a recent publications, that you cite in your manuscript 45“Alternative approaches to target Myc for cancer treatment. Wang C, et al. Signal Transduct Target Ther. 2021. PMID: 33692331 Free PMC article. Review.” They use these terms: The Myc proto-oncogene family consists of three members, C-MYC, MYCN, and MYCL, which encodes the transcription factor c-Myc (hereafter Myc), N-Myc, and L-Myc, respectively. As your article is also a review, I believe that you must have a correct way of citing and that in your current manuscript you could use the most correct form based on official terminology.

Lines 13-20: I didn't find your objective very clear: do you want to explain the MYC pathways to propose new treatments for any cancer? and then you want to discuss the implication of MYC signaling in ovarian cancer carcinogenesis, in the detection of ovarian cancer and still show the therapeutic potential of targeting polyamine metabolism in ovarian cancers? With so many objectives, the text is not linear.

For example, in the lines 33-40 of your introduction you go from c-MYC in mice leading to B cell lymphomagenesis to unapproved drugs. The same occurs for lines 41 to 53.

I believe that the objectives of the abstract and of this beginning of the introduction could be more specific for ovarian cancer since there are many reviews about MYC and cancer in general. And that your review could be systematic focused in ovarian cancer. From line 60 onwards, the text of item 2.1. flows just a little better. Still, there are citations of articles that are vague for the reader. For example: Kohler et al. 1989 (more than 30 years ago) found c-MYC expression in all ovarian tumor tissues, and 10 out of 21 tumors had increased mRNA levels compared to non-malignant tissues. What is the point? The other references, 41 (1993), 46 (1992) are also 30 years old or almost, 39 (a little more recent 2011), and this knowledge is already updated in references 40 (adequate 2018) and 47 (adequate 2020). When going back to articles that are over 30 years old, what is your goal?

From lines 68 to 81 do you show that c-MYC is overexpressed in 35% to 75% of ovarian carcinomas but without clearly stating why these differences and if this is related to cancer carcinogens, promotion, progression?

From lines 93-107 you cite that the studies are not concordant but you do not provide hypotheses.

Lines 141-143 the conclusion is “Despite the above-mentioned studies, c-MYC has largely been considered as “undruggable” for decades, and there are no drugs clinically available for targeting c-142 MYC in MYC-driven cancers. An alternative strategy is to target one or several down-143 stream signaling pathways of c-MYC” but it is not clear to the reader what the relationship is established between what was mentioned above and the use of to target one or several down-stream signaling pathways of c-MYC.

From line 146 to 212 the text is clearer

From line 213 to 244 again gets confused. What is the purpose of these paragraphs?

Figure 1 is beautiful but I didn't find enough background in the text to build this model of Biosynthesis, metabolism, transport and targeting of polyamines. Even the figure is not cited in the text.

From lines 275 to 321 the authors talk about polyamines, their catabolism and cancer, and finally the regulation of polyamines by c-MYC. Perhaps this could be the purpose of the manuscript and rewrite the review in light of that purpose.

From line 360 ​​to line 474 studies referring to diagnosis with polyamines are cited and I would do a systematic and not a simple review here, in the same way as from line 475 to 500, in which in just a very short paragraph, the authors address the perspectives of polyamines in the treatment of ovarian cancer. In a previous article Fahrmann, J.F.; Irajizad, E.; Kobayashi, M.; Vykoukal, J.; Dennison, J.B.; Murage, E.; Wu, R.; Long, J.P.; Do, K.A.; Celestino, J.; et 566 al. A MYC-Driven Plasma Polyamine Signature for Early Detection of Ovarian Cancer. Cancers (Basel) 2021, 13, 567 doi:10.3390/cancers13040913. The authors showed their results on the topic and concluded that a MYC-driven plasma polyamine signature associated with OvCa that complemented CA125 in detecting early-stage ovarian cancer.

And I'm not sure I would end the conclusion with the sentence "Targeting polyamine metabolism in MYC-driven cancers, including ovarian cancer, is a viable strategy that warrants further exploration (line 517-518).

Author Response

It would be useful for the reader to keep the same form of abbreviation. In your text we find this: Line 9: MYC family members, including c-MYC, MYCN, MYCL and Line 29: The MYC family of transcription factors C-MYC, N-MYC and L-MYC. In a recent publication, that you cite in your manuscript 45“Alternative approaches to target Myc for cancer treatment. Wang C, et al. Signal Transduct Target Ther. 2021. PMID: 33692331 Free PMC article. Review.” They use these terms: The Myc proto-oncogene family consists of three members, C-MYC, MYCN, and MYCL, which encodes the transcription factor c-Myc (hereafter Myc), N-Myc, and L-Myc, respectively. As your article is also a review, I believe that you must have a correct way of citing and that in your current manuscript you could use the most correct form based on official terminology.

Response: We have now amended to the article for consistency in our nomenclature of c-MYC as well as its paralogues MYCL and MYCN.

Lines 13-20: I didn't find your objective very clear: do you want to explain the MYC pathways to propose new treatments for any cancer? and then you want to discuss the implication of MYC signaling in ovarian cancer carcinogenesis, in the detection of ovarian cancer and still show the therapeutic potential of targeting polyamine metabolism in ovarian cancers? With so many objectives, the text is not linear.

For example, in the lines 33-40 of your introduction you go from c-MYC in mice leading to B cell lymphomagenesis to unapproved drugs. The same occurs for lines 41 to 53.

Response: We have modified the manuscript to focus on the role of c-MYC and its paralogues in ovarian cancer, c-MYC mediated regulation of polyamine metabolism, targeting of polyamine metabolism via small molecule inhibitors for treatment of ovarian cancer, and the utility of polyamines in biofluids for detecting ovarian cancers.

I believe that the objectives of the abstract and of this beginning of the introduction could be more specific for ovarian cancer since there are many reviews about MYC and cancer in general. And that your review could be systematic focused in ovarian cancer. From line 60 onwards, the text of item 2.1. flows just a little better. Still, there are citations of articles that are vague for the reader. For example: Kohler et al. 1989 (more than 30 years ago) found c-MYC expression in all ovarian tumor tissues, and 10 out of 21 tumors had increased mRNA levels compared to non-malignant tissues. What is the point? The other references, 41 (1993), 46 (1992) are also 30 years old or almost, 39 (a little more recent 2011), and this knowledge is already updated in references 40 (adequate 2018) and 47 (adequate 2020). When going back to articles that are over 30 years old, what is your goal?

Response: We have modified the abstract to better describe the purpose of the review. We have additionally removed selected older studies.

From lines 68 to 81 do you show that c-MYC is overexpressed in 35% to 75% of ovarian carcinomas but without clearly stating why these differences and if this is related to cancer carcinogens, promotion, progression?

Response: We have modified the corresponding text to now state that “c-MYC transcript and protein levels are higher in ovarian tumors compared to normal tissues, and c-MYC overexpression is associated with more aggressive disease.”

From lines 93-107 you cite that the studies are not concordant but you do not provide hypotheses.

Lines 141-143 the conclusion is “Despite the above-mentioned studies, c-MYC has largely been considered as “undruggable” for decades, and there are no drugs clinically available for targeting c-142 MYC in MYC-driven cancers. An alternative strategy is to target one or several down-143 stream signaling pathways of c-MYC” but it is not clear to the reader what the relationship is established between what was mentioned above and the use of to target one or several down-stream signaling pathways of c-MYC.

Response: We have now better clarified the text.

“Despite the above-mentioned studies, targeting c-MYC and its paralogues has remained challenging due to protein and peptides inhibitors undergoing rapid proteolytic degradation, as well as issues related to low membrane and cell permeability, low oral bioavailability, high clearance, and poor tissue distribution.[81] Currently, there are no drugs clinically available for targeting c-MYC in human cancers. An alternative strategy is to target one or several downstream signaling pathways of c-MYC.”

From line 146 to 212 the text is clearer From line 213 to 244 again gets confused. What is the purpose of these paragraphs?

Response: In addition to de novo polyamine biosynthesis, cancer cell can scavenge extracellular polyamines to promote cancer cell growth. Targeting of polyamine transport machinery is therefore a viable therapeutic strategy that is being actively explored (e.g. see content related to AMXT 1501). Consequently, we feel it is pertinent to include content related to mechanisms of polyamine uptake and efflux.

Figure 1 is beautiful but I didn't find enough background in the text to build this model of Biosynthesis, metabolism, transport and targeting of polyamines. Even the figure is not cited in the text.

From lines 275 to 321 the authors talk about polyamines, their catabolism and cancer, and finally the regulation of polyamines by c-MYC. Perhaps this could be the purpose of the manuscript and rewrite the review in light of that purpose.

Response: We thank the Reviewer for catching this oversight. We have restructured the review to better connect the sources of polyamines and their catabolism in ovarian cancer, regulation of polyamines by c-MYC, therapeutic potential of targeting polyamine metabolism in ovarian cancer, as well as the highlight utility for polyamines in biofluids for early detection applications and for potentially selecting individuals who may best benefit from polyamine targeting therapies.

From line 360 to line 474 studies referring to diagnosis with polyamines are cited and I would do a systematic and not a simple review here, in the same way as from line 475 to 500, in which in just a very short paragraph, the authors address the perspectives of polyamines in the treatment of ovarian cancer. In a previous article Fahrmann, J.F.; Irajizad, E.; Kobayashi, M.; Vykoukal, J.; Dennison, J.B.; Murage, E.; Wu, R.; Long, J.P.; Do, K.A.; Celestino, J.; et 566 al. A MYC-Driven Plasma Polyamine Signature for Early Detection of Ovarian Cancer. Cancers (Basel) 2021, 13, 567 doi:10.3390/cancers13040913. The authors showed their results on the topic and concluded that a MYC-driven plasma polyamine signature associated with OvCa that complemented CA125 in detecting early-stage ovarian cancer.

Response: We have modified the respective section to now provide a more systematic review.

And I'm not sure I would end the conclusion with the sentence "Targeting polyamine metabolism in MYC-driven cancers, including ovarian cancer, is a viable strategy that warrants further exploration (line 517-518).

Response: We appreciate the Reviewer’s point of view, and we have modified the conclusion per their recommendation.

Reviewer 3 Report

The authors systematic summarized the polyamine metabolism was involved in the initiation and progression of MYC-driven cancers including ovarian cancer. MYC-driven polyamines and their acetylated derives have been a potential biomarker and therapeutic targeting for ovarian cancer. The manuscript is very interesting. These literatures were integrated to provide a better understanding for the anti-tumor of the synthetic polyamine analogues in ovarian cancer. In my opinion, some minor revision should be resolved before accepted for publication.

1.     Please carefully check SSAT/APOX pathways in Line 288, is it correct to SSAT/PAOX pathway?

2.     In 4.4.2 exploiting synthetic polyamine analogues for ovarian cancer treatment, these analogues have been pronounced anti-cancer effects, which could cooperate with downregulated c-MYC function?

Author Response

  1. Please carefully check SSAT/APOX pathways in Line 288, is it correct to SSAT/PAOX pathway?

Response: We thank the Reviewer for catching this error and we have now made the correction.

  1. In 4.4.2 exploiting synthetic polyamine analogues for ovarian cancer treatment, these analogues have been pronounced anti-cancer effects, which could cooperate with downregulated c-MYC function?

Response: This is an intriguing aspect and worth investigation. To-date, we are not aware of any study in ovarian cancer that has evaluated whether dual targeting polyamine and c-MYC results in improved anti-cancer effects.